# Mendelian Randomization Analyses of Chronic Immune-Mediated Diseases, Circulating Inflammatory Biomarkers, and Cytokines in Relation to Liver Cancer

**DOI:** 10.3390/cancers15112930

**Published:** 2023-05-26

**Authors:** Qiushi Yin, Qiuxi Yang, Wenjie Shi, Ulf D. Kahlert, Zhongyi Li, Shibu Lin, Qifeng Song, Weiqiang Fan, Li Wang, Yi Zhu, Xiaolong Huang

**Affiliations:** 1Department of Hepatobiliary Surgery, The First Affiliated Hospital of Hainan Medical University, Haikou 570100, China; yqsmyth@163.com (Q.Y.); yqx19830323@163.com (Q.Y.); shibulin_medical@126.com (S.L.); qifengsong_hainan@163.com (Q.S.); fanweiqiang2013@126.com (W.F.); lw8849@163.com (L.W.); 2Molecular and Experimental Surgery, University Clinic for General-, Visceral-, Vascular- and Trans-Plantation Surgery, Medical Faculty University Hospital Magdeburg, Otto-von Guericke University, 39120 Magdeburg, Germany; wenjie.shi@uni-oldenburg.de (W.S.); ulf.kahlert@med.ovgu.de (U.D.K.); 3Department of General, Visceral, and Transplant Surgery, Ludwig-Maximilians-University Munich, 80539 Munich, Germany; zhongyi.li@med.uni-muenchen.de; 4Department of Gastroenterological Surgery, The Affiliated Hospital of Jiaxing University, Jiaxing 314001, China

**Keywords:** liver cancer, immune-mediated disease, inflammatory biomarkers, inflammatory cytokines

## Abstract

**Simple Summary:**

Liver cancer is a prevalent gastrointestinal carcinoma and is closely linked to chronic inflammation, including both hepatic and extrahepatic inflammations. However, the genetic association between inflammatory traits and liver cancer has not been systematically investigated. In this study, we aimed to explore the potential causal associations between immune-mediated diseases, circulating inflammatory biomarkers and cytokines, and liver cancer using Mendelian randomization (MR) analysis. To our best knowledge, this is the most comprehensive MR study on this topic to date, involving more than 200 inflammatory traits. This is an important contribution to the field, as it provides insights into the potential causal inflammatory factors of liver cancer.

**Abstract:**

Liver cancer is closely linked to chronic inflammation. While observational studies have reported positive associations between extrahepatic immune-mediated diseases and systemic inflammatory biomarkers and liver cancer, the genetic association between these inflammatory traits and liver cancer remains elusive and merits further investigation. We conducted a two-sample Mendelian randomization (MR) analysis, using inflammatory traits as exposures and liver cancer as the outcome. The genetic summary data of both exposures and outcome were retrieved from previous genome-wide association studies (GWAS). Four MR methods, including inverse-variance-weighted (IVW), MR-Egger regression, weighted-median, and weighted-mode methods, were employed to examine the genetic association between inflammatory traits and liver cancer. Nine extrahepatic immune-mediated diseases, seven circulating inflammatory biomarkers, and 187 inflammatory cytokines were analyzed in this study. The IVW method suggested that none of the nine immune-mediated diseases were associated with the risk of liver cancer, with odds ratios of 1.08 (95% CI 0.87–1.35) for asthma, 0.98 (95% CI 0.91–1.06) for rheumatoid arthritis, 1.01 (95% CI 0.96–1.07) for type 1 diabetes, 1.01 (95% CI 0.98–1.03) for psoriasis, 0.98 (95% CI 0.89–1.08) for Crohn’s disease, 1.02 (95% CI 0.91–1.13) for ulcerative colitis, 0.91 (95% CI 0.74–1.11) for celiac disease, 0.93 (95% CI 0.84–1.05) for multiple sclerosis, and 1.05 (95% CI 0.97–1.13) for systemic lupus erythematosus. Similarly, no significant association was found between circulating inflammatory biomarkers and cytokines and liver cancer after correcting for multiple testing. The findings were consistent across all four MR methods used in this study. Our findings do not support a genetic association between extrahepatic inflammatory traits and liver cancer. However, larger-scale GWAS summary data and more genetic instruments are needed to confirm these findings.

## 1. Introduction

Liver cancer is a common digestive system malignancy, with approximately 906,000 new cases and 830,000 deaths reported worldwide in 2020 [1]. It has been determined that liver cancer is derived from sustained hepatic inflammation caused by a suite of factors including viral hepatitis, alcohol consumption, and/or fatty liver disease [2]. Moreover, mounting evidence has suggested that extrahepatic chronic inflammations also increase the risk of liver cancer. For instance, previous epidemiological studies have reported a positive association between periodontitis and the risk of liver cancer [3,4]. A population-based cohort study showed that psoriasis, psoriatic arthritis, and rheumatoid arthritis were associated with an increased risk of liver cirrhosis [5]. Based on the UK Biobank cohort, He et al. reported that inflammatory bowel disease and its subtypes Crohn’s disease and ulcerative colitis are significantly associated with an elevated risk of liver cancer [6]. On the other hand, in addition to liver enzymes, numerous blood inflammatory biomarkers have been found to be associated with liver cancer risk. For example, Zhu et al. found that serum levels of C-reactive protein (CRP) are associated with the risk of liver cancer in a dose–response manner [7]. A similar positive association was observed between levels of IL6 and liver cancer [8]. These findings suggested that chronic inflammation is closely involved in hepatic tumorigenesis.

Despite mounting evidence from previous observational studies, it is hard to conclude that chronic immune-mediated diseases and inflammatory biomarkers are causal with the onset of liver cancer because of potential unmeasured confounders or reverse causality in observational studies. The inherent pitfalls of observational studies to some extent impede a full understanding of the association between chronic extrahepatic inflammation and liver cancer, which merits further investigations from other perspectives. Mendelian randomization (MR) analysis that leverages genetic information can serve as a valuable complement to observational studies [9] and has been widely used to explore the causal associations between exposures and diseases [10,11,12,13].

To date, many MR analyses have been performed to assess the association between inflammatory biomarkers and diseases [11,14,15,16]. However, only a few MR analyses have been conducted to assess the association between inflammation and liver cancer [7,17]. Moreover, these MR studies considered only a limited number of inflammatory biomarkers. Given the close relationship between chronic inflammations and liver cancer, it is necessary to systematically examine their impact on liver cancer. To this end, in the current study, we applied two-sample MR methods to assess the genetic associations of ten extrahepatic immune-mediated diseases, seven circulating inflammatory biomarkers (e.g., CRP and leukocyte count), and 228 blood inflammatory cytokines with the risk of liver cancer. Our findings not only provide important complementary information to previous epidemiological studies but also offer novel insights into the pathogenesis of liver cancer.

## 2. Methods

### 2.1. Study Design

We conducted a two-sample MR analysis, where the extrahepatic immune-mediated diseases were considered primary exposures, and circulating inflammatory biomarkers and cytokines were secondary exposures. The outcome of interest was liver cancer. For this analysis, we utilized GWAS summary data of the exposures and the outcome from study populations with the same ethnic background (i.e., Europeans), but without any overlap in individuals, to ensure their independence.

### 2.2. GWAS of Exposures

We examined ten immune-mediated diseases in this study, namely asthma [18], rheumatoid arthritis [19], type 1 diabetes [20], psoriasis [21], Crohn’s disease [22], ulcerative colitis [22], celiac disease [23], multiple sclerosis [24], systemic lupus erythematosus [25], and periodontitis [26]. We also retrieved GWAS summary data of seven circulating inflammatory biomarkers, including CRP [27], leukocyte count, eosinophil count, basophil count, neutrophil count, lymphocyte count, and monocyte count [28]. The details of the selected GWAS for the exposures are presented in Appendix A. All selected GWASs were conducted on individuals of European ancestry, with large sample sizes, and quality control procedures were implemented. Further information on these GWASs can be found in the respective studies.

For circulating inflammatory cytokines, we retrieved the GWAS summary data from a plasma proteome GWAS study that involved 4907 aptamers in 35,559 Icelanders [29]. All plasma samples were measured with the SomaScan version 4 assay (SomaLogic), which contains 5284 aptamers providing a measurement of the relative binding of the plasma sample to each of the aptamers in relative fluorescence units. The genotype information was derived from Illumina SNP chips, long-range phased, and imputed based on the sequenced dataset. After quality control processes, 27.2 million imputed variants with minor allele frequency (MAF) > 0.01% and imputation information > 0.9 were analyzed in GWAS. Each of the 4907 aptamers that were tested underwent rank-inverse normalization, with adjustment for age, sex, and sample age for both the deCODE Health study and the remaining studies. The resulting residuals were then standardized again using rank-inverse normalization and used as phenotypes for genome-wide association testing via a linear mixed model implemented in BOLT-LMM. Our study retrieved GWAS summary data of 228 inflammatory cytokines, including 37 chemokines, 82 interleukins, 44 growth factors, 22 interferons, 37 tumor necrosis factors (TNF), and 6 other types.

### 2.3. GWAS of Outcome

The summary genetic statistics of liver cancer were retrieved from the FinnGen research project (https://r7.finngen.fi/, accessed on 20 January 2023; Version R7). FinnGen is a public–private partnership project combining genotype data from Finnish biobanks and digital health record data from Finnish health registries. The GWAS for liver cancer, defined as malignant neoplasm of the liver and intrahepatic bile ducts in the FinnGen study, included 518 cases and 238,678 controls without any type of cancer. More information about the GWAS in the FinnGen study can be found on their website (https://finngen.gitbook.io/documentation/, accessed on 20 January 2023). Briefly, DNA samples were genotyped with Illumina (Illumina Inc., San Diego, CA, USA) and Affymetrix arrays (Thermo Fisher Scientific, Santa Clara, CA, USA). In sample-wise quality control steps, individuals with ambiguous gender, high genotype missingness (>5%), excess heterozygosity, and non-Finnish ancestry were excluded. In variant-wise quality control steps, variants with high missingness (>2%), low HWE *p*-value (<1 × 10^−6^), and low minor allele count (<3) were excluded. Age, sex, 10 principal components, and FinnGen 1 or 2 chip or legacy genotyping batch were used as covariates in the GWAS, which was implemented using Regenie software (V2.2.4).

### 2.4. Mendelian Randomization Analysis

#### 2.4.1. Selection of Instrumental Variables

We used a multistep process to select the genetic instrumental variables (IVs). First, we extracted SNPs that were associated with the exposures at the conventional genome-wide association study (GWAS) threshold (*p* < 5 × 10^−8^). Next, we clumped the SNPs based on linkage disequilibrium (LD) estimates from the European samples in the 1000 Genomes project, using an LD threshold of *R*^2^ < 0.01 and a window size of 10,000 kb. We then extracted the corresponding beta coefficients and standard errors of the selected SNPs from the GWAS of liver cancer. For SNPs that were not present in the GWAS of liver cancer, we retrieved data on an SNP proxy with an LD estimate of *R*^2^ > 0.8 with the requested SNP. Finally, we corrected or excluded ambiguous SNPs with inconsistent alleles and palindromic SNPs with ambiguous strands. To ensure the reliability of IVs, we calculated the F-statistics to assess the strength of the relationship between IVs and phenotype using the following equation [30]:$$F=\frac{R^{2}/k}{\left(1-R^{2}\right)/\left(n-k-1\right)}$$
where *R*^2^ is the proportion of phenotype that can be explained by the genetic information, *k* is the number of instruments used in the model, and *n* is the sample size. An F-statistic > 10 indicates the suitability of the IVs, namely, meeting the first assumption of MR analysis [31].

#### 2.4.2. Statistical Analysis and Sensitivity Analysis

We conducted a two-sample MR analysis using the following steps to investigate the potential causal relationship between immune-mediated diseases and circulating inflammatory biomarkers and liver cancer [32]: (1) harmonizing the exposure data and outcome data by matching the SNPs; (2) using the inverse-variance-weighted (IVW) method to test for between-SNP heterogeneity, with a *p* value greater than 0.05 for the Q-statistic indicating an absence of heterogeneity; (3) employing the MR-Egger regression intercept test to identify horizontal pleiotropy; (4) using the IVW method to examine the genetic association between exposure and outcome. We also conducted sensitivity analyses using MR-Egger regression, weighted-median, and weighted-mode methods. The MR-Egger regression is based on the InSIDE (INstrument Strength Independent of Direct Effect) assumption and consists of three parts: (i) a test for directional pleiotropy, (ii) a test for a causal effect, and (iii) an estimate of the causal effect [33]. The weighted-median and weighted-mode methods are more robust than IVW and MR-Egger methods when over 50% of SNPs are invalid instruments [34,35]. We also calculated the statistical power for MR analysis using the mRnd website (https://shiny.cnsgenomics.com/mRnd/, accessed on 20 March 2023) [36]. 

For inflammatory cytokines, we first assessed their relationship with liver cancer using the IVW method with multiple IVs or the Wald ratio test with only one IV. Cytokines that showed significant associations with liver cancer after correcting for multiple testing were further validated using MR-Egger regression, weighted-median, and weighted-mode methods. To validate the results, we performed a repeated analysis using GWAS of liver cancer from the UK Biobank, in which 539 cases and 419,992 controls were included. (https://pan.ukbb.broadinstitute.org/phenotypes/index.html, accessed on 16 May 2023). 

All statistical analyses were performed using the R program (v4.1.1). MR analysis was performed using *TwoSampleMR* and *MendelianRandomization* packages. The Bonferroni method was employed to correct for multiple testing.

## 3. Results

### 3.1. Association between Immune-Mediated Diseases and Liver Cancer

In this study, we utilized a different number of IVs for each immune-mediated disease, with 225, 117, 131, 84, 106, 76, 11, 55, and 48 IVs used for asthma, rheumatoid arthritis, type 1 diabetes, psoriasis, Crohn’s disease, ulcerative colitis, celiac disease, multiple sclerosis, and systemic lupus erythematosus, respectively (Table 1). We excluded periodontitis from the analysis due to the unavailability of valid IVs. All nine exposures had mean-F statistics greater than 10, suggesting a low probability of weak IV bias. Additionally, there was no between-SNP heterogeneity or horizontal pleiotropy detected for any of the exposures using the IVW method or the MR-Egger regression intercept test (Table 1). The statistical power was greater than 95% for detecting an odds ratio (OR) less than 0.9 or greater than 1.1, which decreased to 24–88% when identifying an OR between 0.9 and 1.1.

According to the IVW method, there was no significant association between any of the nine immune-mediated diseases and the risk of liver cancer. The OR estimates were as follows: 1.08 (95% CI 0.87–1.35) for asthma, 0.98 (95% CI 0.91–1.06) for rheumatoid arthritis, 1.01 (95% CI 0.96–1.07) for type 1 diabetes, 1.01 (95% CI 0.98–1.03) for psoriasis, 0.98 (95% CI 0.89–1.08) for Crohn’s disease, 1.02 (95% CI 0.91–1.13) for ulcerative colitis, 0.91 (95% CI 0.74–1.11) for celiac disease, 0.93 (95% CI 0.84–1.05) for multiple sclerosis, and 1.05 (95% CI 0.97–1.13) for systemic lupus erythematosus (Figure 1). The results obtained using the other three MR methods, namely MR-Egger regression, weighted-median, and weighted-mode, were consistent with those of the IVW method. Figure 2 displays the scatter plots depicting the SNP effects on both the exposures and the outcome.

### 3.2. Association between Circulating Inflammatory Biomarkers and Liver Cancer

Table 1 displays the number of instrumental variables (IVs) used for each inflammatory biomarker in the MR analysis: 291 for CRP, 185 for leukocyte count, 208 for eosinophil count, 83 for basophil count, 162 for neutrophil count, 193 for lymphocyte count, and 266 for monocyte count. We observed no evidence of weak-IV bias based on the mean F-statistics, and there was no significant horizontal pleiotropy for any of the exposures. However, we detected significant between-SNP heterogeneity for CRP (*p* = 0.002) and neutrophil count (*p* = 0.038). The statistical power was high, with >70% power to detect an odds ratio (OR) between 0.9 and 1.1 and >95% power to detect an OR > 1.1 or <0.9.

Our MR analysis did not reveal any significant associations between circulating inflammatory biomarkers and liver cancer, with an OR of 1.13 (95% CI 0.81–1.57) for CRP, 0.82 (95% CI 0.59–1.14) for leukocyte count, 0.82 (95% CI 0.63–1.07) for eosinophil count, 1.53 (95% CI 0.93–2.51) for basophil count, 0.80 (95% CI 0.56–1.15) for neutrophil count, 0.81 (95% CI 0.59–1.10) for lymphocyte count, and 0.93 (95% CI 0.74–1.16) for monocyte count (Figure 3). The results were consistent across the other three MR methods. Figure 4 displays scatter plots of SNP effects on both exposures and outcomes.

### 3.3. Association between Circulating Inflammatory Cytokines and Liver Cancer

After performing quality control, we included 187 inflammatory cytokines (30 chemokines, 72 interleukins, 31 growth factors, 17 interferons, 31 TNFs, and 6 others) for MR analysis (Appendix A). Among these, 166 cytokines had two or more valid genetic variants, while the remaining 21 cytokines only had one valid IV. We observed significant associations between the risk of liver cancer and three cytokines: interleukin-23 receptor (OR = 0.60, 95% CI 0.38–0.95, *p* = 0.028), interleukin-27 receptor subunit alpha (OR = 1.16, 95% CI 1.01–1.13, *p* = 0.036), and C-X-C motif chemokine 16 (OR = 0.78, 95% CI 0.61–1.00, *p* = 0.046) (Figure 5; Appendix A). However, these associations were not statistically significant after Bonferroni correction for multiple testing (0.05/187). The repeated MR analysis using GWAS summary data from the UK Biobank yielded consistent results compared to our main results (Appendix A). Although a few suggestive association was detected, for example, CXCL17 (OR = 1.45, 95 CI% 1.06–1.79, *p* = 0.011), no significant association remained after correcting for multiple testing.

## 4. Discussion

In this study, we conducted a comprehensive analysis of the genetic association between a range of chronic inflammatory diseases, biomarkers, and cytokines with the risk of liver cancer. Our study included nine extrahepatic inflammatory diseases, seven circulating inflammatory biomarkers, and 187 inflammatory cytokines, making it the most comprehensive investigation to date on the impact of systemic inflammation on liver cancer. Although we found three cytokines that showed a potential association with liver cancer at a significance threshold of *p* < 0.05, these associations did not remain significant after correcting for multiple testing using the Bonferroni method. Therefore, the genetic evidence did not support any causal relationship between inflammatory traits and liver cancer risk. These findings suggest that the previously reported correlations from observational studies might be confounded.

Numerous risk factors for liver cancer have been extensively investigated and well-determined [37], which has greatly contributed to the prevention of liver cancer in the general population. However, it is important to note that risk factors are not equivalent to etiologies, and only a few risk factors, such as viral hepatitis and aflatoxin, have been defined as etiologies for liver cancer. Understanding the etiologies for liver cancer is crucial to comprehending the disease pathogenesis and developing cost-effective approaches to prevent the development of this lethal disease. Nevertheless, conventional observational studies, including prospective cohort studies, may have inherent limitations in discovering etiology. Confounding, reverse causation, and various biases can affect the associations to varying degrees, and even with careful study design and statistical adjustment, incorrect causal inference is possible [38]. In comparison, MR analysis has several strengths: it is immune to confounders since genotypes are allocated during meiosis, less affected by information bias as genotype information can be accurately obtained through sequencing, and easy to perform as it only requires GWAS summary data instead of individual data [39]. Furthermore, MR interpretation of a statistically significant association as evidence that the exposure has a causal effect on the outcome is an important feature [40]. In this regard, MR analysis can serve as a valuable complement to observational studies.

Since liver cancer is closely linked to chronic inflammation [41], we applied a set of MR methods to examine the association between inflammatory traits and liver cancer and aimed to determine potential causal inflammatory factors for liver cancer. Previous population-based studies have investigated the association between certain inflammatory traits and liver cancer. For instance, a systematic review and meta-analysis of epidemiological studies reported a 90% increased risk of liver cancer in patients with psoriasis compared to healthy controls [42]. An umbrella meta-analysis reported that Crohn’s disease was associated with a 2.18-fold increased risk of liver cancer [43]. Similar positive associations were observed for type 1 diabetes and systemic lupus erythematosus [44,45]. Kim et al. reported a significantly negative association (multivariable-adjusted hazard ratio = 0.80) between asthma and liver cancer [46]. However, using MR analyses, we did not detect a genetic association between these inflammatory diseases and liver cancer. Given that complex diseases such as liver cancer and Crohn’s disease are partly determined by genetic factors, the null genetic association suggests that there may be no overlap in the genetic architecture between these two conditions.

A similar null association was observed between genetically predicted inflammatory biomarkers and cytokines and liver cancer. For example, our MR analysis did not find a genetic association between genetically predicted CRP and liver cancer, although previous epidemiological studies have identified CRP as an independent risk factor for liver cancer [7,47]. Our MR estimate for CRP was consistent with a previous MR study [7]. Similarly, we found no genetic association between vascular endothelial growth factor (VEGF) and liver cancer, consistent with the results of a study by Wu et al. [17], though this cytokine has been demonstrated to be involved in hepatic tumorigenesis [48]. The lack of association between circulating inflammatory traits and liver cancer suggests that these biomarkers were unlikely to be the cause of liver cancer, but rather acted as response markers to environmental risk factors (such as smoking, alcohol consumption, aging, and obesity), which can induce chronic low-grade inflammation. However, this hypothesis requires further validation because only a small proportion of the variance of the biomarkers can be explained by the genetic instrumental variable [7].

The main strength of our study is that it is the most comprehensive study to date using MR analysis to examine the association between over 200 inflammatory traits and liver cancer. However, we acknowledge several limitations of our study. First, we obtained GWAS summary data from the FinnGen project, which includes only 518 liver cancer cases. While the sample size was large, a small number of cases can limit the statistical power of GWAS, potentially leading to missed genetic signals [39]. We should bear in mind that the MR estimates were largely depended on robust IVs. Unfortunately, there is no large-scale GWAS with a standardized design for liver cancer to date. Second, only participants of European ancestry were included in this study, which compromise the generalization of our results to other ancestry populations. Third, the associations between inflammatory traits and liver cancer may be varied across liver cancer etiologies. Due to the lack of GWAS summary data for etiology-specific liver cancer, we did not assess the influence of liver cancer etiologies in this study. Moreover, we cannot assess the associations between exposures and liver cancer according to its histological subtypes due to the data unavailability. Finally, our estimates might also be subject to the inherent shortcomings of MR analysis such as selection bias [49]. Genetic variants which are related to specific phenotypes might also be related to participation [50]. As such, individuals at high genetic risk for inflammatory diseases may be more likely to drop out of the cohort due to a higher susceptibility to these conditions than those at low genetic risk.

## 5. Conclusions

To conclude, our comprehensive MR analysis did not reveal any evidence of a causal effect between genetically predicted immune-mediated diseases and circulating inflammatory biomarkers and cytokines and liver cancer. However, due to the limitations of our study, including a relatively small number of liver cancer cases and the use of only European ancestry populations, our findings should be confirmed by further studies using larger-scale GWAS summary data and more genetic instruments.

## Figures and Tables

**Figure 1 cancers-15-02930-f001:**
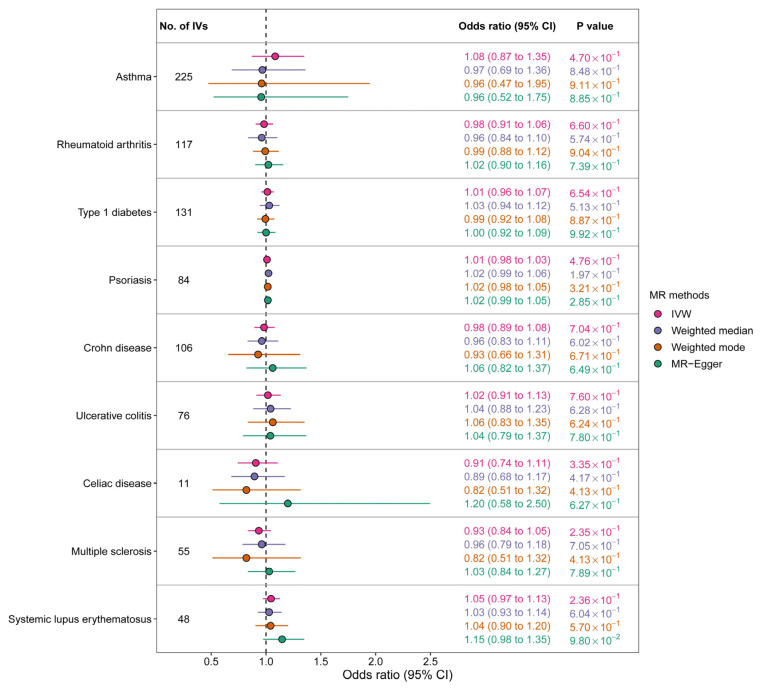
Genetic association between immune-mediated diseases and liver cancer according to Mendelian randomization analysis (IV, instrumental variable; IVW, inverse-variance-weighted method).

**Figure 2 cancers-15-02930-f002:**
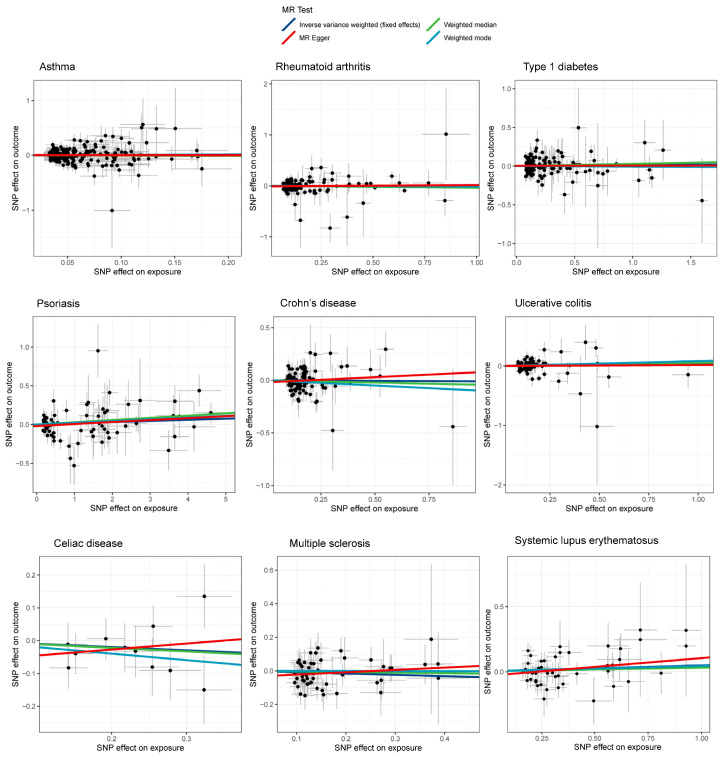
Scatter plot showing the SNP effects on both immune-mediated diseases and liver cancer (The gray error bars denote the 95% confidence intervals of the effects).

**Figure 3 cancers-15-02930-f003:**
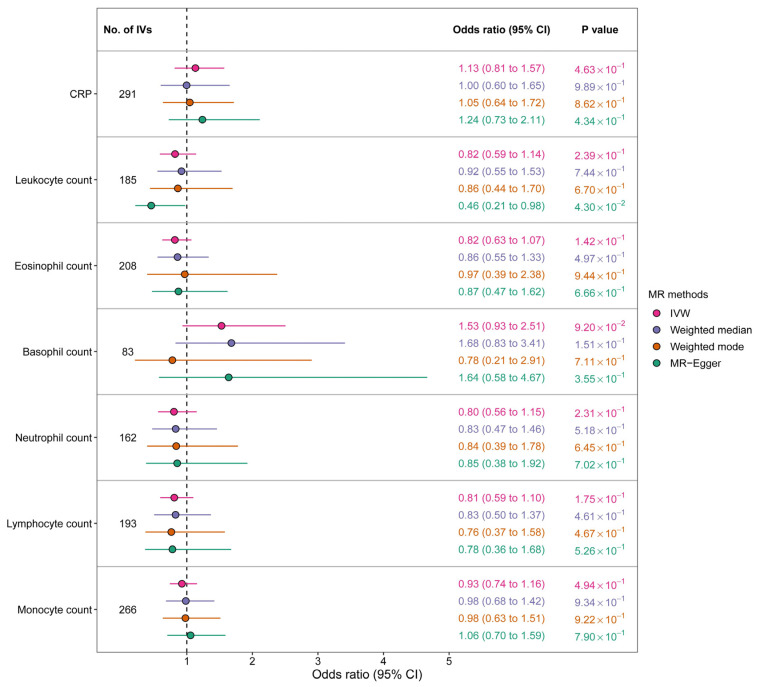
Genetic association between circulating inflammatory biomarkers and liver cancer according to Mendelian randomization analysis (CRP, C-reactive protein; IV, instrumental variable; IVW, inverse-variance-weighted method).

**Figure 4 cancers-15-02930-f004:**
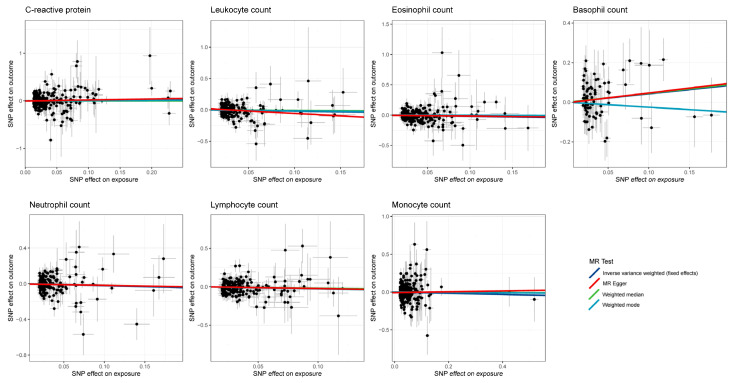
Scatter plot showing the SNP effects on both circulating inflammatory biomarkers and liver cancer (The gray error bars denote the 95% confidence intervals of the effects).

**Figure 5 cancers-15-02930-f005:**
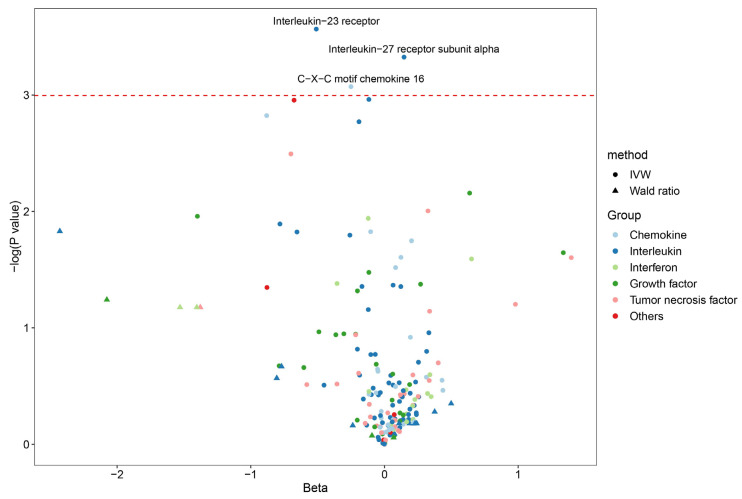
Genetic association between circulating inflammatory cytokines and liver cancer according to Mendelian randomization analysis (We only show the point estimate in this plot. The red dotted line denotes the threshold of *p* value < 0.05).

**Table 1 cancers-15-02930-t001:** Statistics of Mendelian randomization analysis for immune-mediated diseases and liver cancer.

Exposures		No. of IV	F-Statistics	Between-SNP Heterogeneity		Horizontal Pleiotropy		Statistical Power to Detect OR <0.9 or >1.1 (%)	Statistical Power to Detect OR between 0.9 and 1.1 (%)
				Q-Value	*p* Value	Egger- Intercept	*p* Value		
Immune-mediated diseases	Asthma	225	588.9	217.8	0.453	0.0043	0.785	100	88
	Rheumatoid arthritis	117	98.5	114.4	0.470	−0.0093	0.431	97	24
	Type 1 diabetes	131	674.5	112.0	0.809	0.0038	0.732	100	80
	Psoriasis	84	255.6	79.7	0.104	−0.019	0.331	99	41
	Crohn’s disease	106	322.1	100.4	0.306	−0.0189	0.385	100	64
	Ulcerative colitis	76	98.5	73.1	0.284	0.0019	0.936	100	84
	Celiac disease	11	21.5	7.7	0.655	−0.0635	0.465	100	32
	Multiple sclerosis	55	266.8	42.3	0.852	−0.0361	0.217	100	55
	Systemic lupus erythematosus	48	198.9	54.0	0.225	−0.0352	0.218	100	62
Circulating inflammatory biomarkers	C-reactive protein	291	458.9	365.2	0.002	−0.0036	0.674	100	85
	Leukocyte count	185	225.3	209.1	0.099	0.0224	0.095	100	87
	Eosinophil count	208	198.5	203.5	0.556	−0.0028	0.822	100	90
	Basophil count	83	110.3	97.8	0.112	−0.0027	0.883	100	71
	Neutrophil count	162	196.6	194.3	0.038	−0.0024	0.871	100	80
	Lymphocyte count	193	288.3	186.8	0.592	0.0013	0.922	100	84
	Monocyte count	266	300.7	282.3	0.223	−0.0070	0.447	100	92

IV, instrumental variables; OR, odds ratio.

## Data Availability

The datasets analyzed during the current study are available in Appendix A.

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
