# Peer review of "Mendelian Randomization Analyses of Chronic Immune-Mediated Diseases, Circulating Inflammatory Biomarkers, and Cytokines in Relation to Liver Cancer"

_cancers, 2023, doi:10.3390/cancers15112930_

Round 1

Reviewer 1 Report

Thank you for the opportunity to review this article. In the current article, Yin et al aimed to explore the potential causal associations between immune-mediated diseases, circulating inflammatory biomarkers and cytokines, and liver cancer using Mendelian randomization (MR) analysis. They included a total of ten extrahepatic immune-mediated diseases, seven circulating inflammatory biomarkers, and 228 blood inflammatory cytokines as the exposures. Overall, this is an interesting and comprehensive study and owns its implications for clinical practice. The language is well written and the figures and tables are well structured. We have some suggestions for this work to improve its qualification to publish in Cancers.

1) Although there was yet no well-designed GWAS for liver cancer, the authors should replicate their analysis using the GWAS of liver cancer from another population, such as the UKBB. The repeating analyses should be explicitly stated in the main results and be detailed in the Supplement.

2) Liver cancer is a to some extent heterogeneous carcinoma, including several histological subtypes. The authors, if it is possible, should perform a subgroup analysis. For example, examining the association between exposures and hepatocellular carcinoma.

3) In figures 1 and 3, a blank background may be better than a gray background. In figure 5, the MR methods used for each cytokine should be noted using shape of the points.

4) Although MR analysis is a good complement for observational studies, it is also susceptible a set of biases such as selection bias. These pitfalls should be explicitly stated in the Discussion part. More importantly, the authors should note that the estimates of MR analysis are largely depend on the genetic data used (ie, IVs).

5) Is there GWAS summary data for other inflammatory biomarkers, such as TNF?

Author Response

###reviewer 1

Thank you for the opportunity to review this article. In the current article, Yin et al aimed to explore the potential causal associations between immune-mediated diseases, circulating inflammatory biomarkers and cytokines, and liver cancer using Mendelian randomization (MR) analysis. They included a total of ten extrahepatic immune-mediated diseases, seven circulating inflammatory biomarkers, and 228 blood inflammatory cytokines as the exposures. Overall, this is an interesting and comprehensive study and owns its implications for clinical practice. The language is well written and the figures and tables are well structured. We have some suggestions for this work to improve its qualification to publish in Cancers.

Response: Thank you very much for the reviewer’s encouraging and helpful comments.

1) Although there was yet no well-designed GWAS for liver cancer, the authors should replicate their analysis using the GWAS of liver cancer from another population, such as the UKBB. The repeating analyses should be explicitly stated in the main results and be detailed in the Supplement.

Response: Thank you for your feedback. In response to the reviewer's request, we have conducted a repeated Mendelian randomization (MR) analysis utilizing the GWAS data of liver cancer derived from the UK Biobank. The details of this analysis can be found on pages 4 and 11 of our revised manuscript, and we have also included Supplementary Table S4 to provide additional information on the results. We believe that this additional analysis strengthens our study and provides valuable insights into the relationship between inflammatory traits and liver cancer.

2) Liver cancer is a to some extent heterogeneous carcinoma, including several histological subtypes. The authors, if it is possible, should perform a subgroup analysis. For example, examining the association between exposures and hepatocellular carcinoma.

Response: Thank you for your review. We appreciate your feedback and totally agree with you that the association between inflammatory traits and liver cancer may be influenced by the different histological subtypes of liver cancer. Regrettably, to date, no GWAS has been conducted for histology-specific liver cancer. We have acknowledged this limitation in our study and made the necessary revisions to our manuscript. Please refer to the track-changes on page 14 for more details.

3) In figures 1 and 3, a blank background may be better than a gray background. In figure 5, the MR methods used for each cytokine should be noted using shape of the points.

Response: Thank you. Per the reviewer’s kind request, we have modified the figures 1, 3, and 5. Please see the updated figures on pages 7, 10, and 12.

4) Although MR analysis is a good complement for observational studies, it is also susceptible a set of biases such as selection bias. These pitfalls should be explicitly stated in the Discussion part. More importantly, the authors should note that the estimates of MR analysis are largely depend on the genetic data used (ie, IVs).

Response: Thank you. We agree with the reviewer. We have discussed the potential limitations of MR analysis and also emphasized the importance of IVs for MR analysis. Please see page 14.

5) Is there GWAS summary data for other inflammatory biomarkers, such as TNF?

Response: Thank you for your comment. Prior to conducting our analysis, we performed a comprehensive literature search to identify any available GWAS studies specifically investigating inflammatory traits beyond those included in our study. However, unfortunately, we did not come across any eligible GWAS that examined inflammatory biomarkers not already included in our analysis. We believe that despite this limitation, our study still provides valuable insights into the association between the inflammatory traits included in our analysis and liver cancer risk.

Reviewer 2 Report

The article is devoted to the actual issue of identifying risk factors for the development of hepatocarcinoma. The fact that the result of the hypothesis test turned out to be negative is also a valuable result. It might be interesting to include in the study a test of the relationship between cancer development and known risk factors for cancer development (hepatitis viruses, etc.) - to check what estimates would be obtained by this method on these data. 

Small recommendations:

The second sentence in the introduction looks like there is no alternative - as if only these three causes cause cancer. I recommend editing it a bit.

Figure captions were above the figures, not below the figure.

Author Response

###reviewer 2

Comment: The article is devoted to the actual issue of identifying risk factors for the development of hepatocarcinoma. The fact that the result of the hypothesis test turned out to be negative is also a valuable result. It might be interesting to include in the study a test of the relationship between cancer development and known risk factors for cancer development (hepatitis viruses, etc.) - to check what estimates would be obtained by this method on these data.

Response: Thank you very much for the reviewer’s encouraging and helpful comments. We appreciate the promising idea proposed by the reviewer. Liver cancer is a complex disease that involves many risk factors such as viral hepatitis, diet, inflammation, and metabolic dysfunction etc. Mendelian randomization analysis has emerged as a valuable tool to investigate the causal relationships between these risk factors and liver cancer risk. For example, in a previous Mendelian randomization study, the authors found that genetic predisposition to smoking initiation was associated with increased risk of liver cancer (Elife. 2023; 12:e84051). Likewise, Deng et al found that alcohol consumption was positively associated with liver cancer risk (Hepatol Commun. 2022;6(8):2147-2154). Kamiza et al revealed that genetic liability to chronic HBV infection causally associated with liver cancer (odds ratio = 1.27, 95% CI 1.20-1.35) (EBioMedicine. 2022; 79:104003).

In the current study, we aimed to assess the genetic association between chronic inflammation and liver cancer. Inspired by the reviewer's insightful suggestion, we plan to conduct a comprehensive Mendelian randomization analysis encompassing a wide spectrum of risk factors for the development of liver cancer in future studies. This approach will enable us to further explore the causal relationships between these risk factors and HCC and expand our understanding of its etiology. We appreciate the reviewer's valuable input and will incorporate these suggestions into our future research endeavors.

Small recommendations:

The second sentence in the introduction looks like there is no alternative - as if only these three causes cause cancer. I recommend editing it a bit.

Response: Thank you. We have revised this sentence according to the reviewer’s suggestion. Please see the first track-change on page 2.

Figure captions were above the figures, not below the figure.

Response: Thank you. We have removed the figure captions to the bottom of figures.

Reviewer 3 Report

The article “Mendelian Randomization Analyses of Chronic Immune-Mediated Diseases, Circulating Inflammatory Biomarkers, and Cytokines in Relation to Liver Cancer.” Liver cancer is the outcome of chronic liver diseases, the majority of which are caused by hepatitis B and hepatitis C virus. Both of these viruses are non-cytopathic, and these viruses are not also directly carcinogenic. The genesis of cancers due to infection with hepatitis B virus (HBV) and hepatitis C virus (HCV) is assumed to follow a path of inflammation and fibrosis. In line with this, observational studies have reported positive associations between extrahepatic immune-mediated diseases and systemic inflammatory biomarkers with hepatocellular carcinoma (HCC). However, after correcting for multiple testing, the authors could not find a significant association between circulating inflammatory biomarkers and cytokines and liver cancer. This seems that a genetic association between extrahepatic provocative traits and liver cancer could not be substantiated, and more studies are needed.

The study is moderately designed and performed accordingly. The point that should be clarified is about the etiologies of cancer subjects. Also, it is necessary to expose if liver cancers due to viral etiologies and other etiological factors are similar or different. These types of analysis using GWAS summary data of the exposures and the outcome is clinically essential.  There is a lack of information about the status of liver cancer analyzed in this study.

Author Response

###reviewer 3

The article “Mendelian Randomization Analyses of Chronic Immune-Mediated Diseases, Circulating Inflammatory Biomarkers, and Cytokines in Relation to Liver Cancer.” Liver cancer is the outcome of chronic liver diseases, the majority of which are caused by hepatitis B and hepatitis C virus. Both of these viruses are non-cytopathic, and these viruses are not also directly carcinogenic. The genesis of cancers due to infection with hepatitis B virus (HBV) and hepatitis C virus (HCV) is assumed to follow a path of inflammation and fibrosis. In line with this, observational studies have reported positive associations between extrahepatic immune-mediated diseases and systemic inflammatory biomarkers with hepatocellular carcinoma (HCC). However, after correcting for multiple testing, the authors could not find a significant association between circulating inflammatory biomarkers and cytokines and liver cancer. This seems that a genetic association between extrahepatic provocative traits and liver cancer could not be substantiated, and more studies are needed.

The study is moderately designed and performed accordingly. The point that should be clarified is about the etiologies of cancer subjects. Also, it is necessary to expose if liver cancers due to viral etiologies and other etiological factors are similar or different. These types of analysis using GWAS summary data of the exposures and the outcome is clinically essential.  There is a lack of information about the status of liver cancer analyzed in this study.

Response: We sincerely appreciate the reviewer's professional and insightful comments on our study. We fully acknowledge and agree with the reviewer's point that the etiology of liver cancer can significantly impact the association between inflammatory traits and the development of liver cancer. We regret to inform that no GWAS specific to etiology-specific liver cancer has been conducted to date. This limitation has been duly acknowledged in our revised manuscript. Please refer to the track-changes on page 14, where the third limitation now explicitly mentions the absence of GWAS studies focused on etiology-specific liver cancer. We value the reviewer's input and are grateful for their contribution to enhancing the clarity and transparency of our research.